# Protocol for DexEnceph: a randomised controlled trial of dexamethasone therapy in adults with herpes simplex virus encephalitis

Thomas Whitfield  ,[1] Cristina Fernandez,[1] Kelly Davies,[2] Sylviane Defres,[1,3,4] Michael Griffiths,[1,5] Cory Hooper,[1] Rebecca Tangney,[6] Girvan Burnside,[7] Anna Rosala- Hallas,[7] Perry Moore,[8] Kumar Das,[9] Mark Zuckerman,[10] Laura Parkes,[11] Simon Keller,[6] Neil Roberts,[12] Ava Easton,[13] Saber Touati,[14] Rachel Kneen,[15,16] J P Stahl,[17] Tom Solomon[18,19]

For numbered affiliations see end of article.

**Correspondence to**
Professor Tom Solomon;
tsolomon@liverpool.ac.uk

## ABSTRACT

**Introduction** Herpes simplex virus (HSV) encephalitis is a rare severe form of brain inflammation that commonly leaves survivors and their families with devastating long-term consequences. The virus particularly targets the temporal lobe of the brain causing debilitating problems in memory, especially verbal memory. It is postulated that immunomodulation with the corticosteroid, dexamethasone, could improve outcomes by reducing brain swelling. However, there are concerns (so far not observed) that such immunosuppression might facilitate increased viral replication with resultant worsening of disease. A previous trail closed early because of slow recruitment.

**Method** DexEnceph is a pragmatic multicentre, randomised, controlled, open-label, observer-blind trial to determine whether adults with HSV encephalitis who receive dexamethasone alongside standard antiviral treatment with aciclovir for have improved clinical outcomes compared with those who receive standard treatment alone. Overall, 90 patients with HSV encephalitis are being recruited from a target of 45 recruiting sites; patients are randomised 1:1 to the dexamethasone or control arms of the study. The primary outcome measured is verbal memory as assessed by the Weschler Memory Scale fourth edition Auditory Memory Index at 26 weeks after randomisation. Secondary outcomes are measured up to 72 weeks include additional neuropsychological, clinical and functional outcomes as well as comparison of neuroimaging findings. Patient safety monitoring occurs throughout and includes the detection of HSV DNA in cerebrospinal fluid 2 weeks after randomisation, which is indicative of ongoing viral replication. Innovative methods are being used to ensure recruitment targets are met for this rare disease.

**Discussion** DexEnceph aims to be the first completed randomised controlled trial of corticosteroid therapy in HSV encephalitis. The results will provide evidence for future practice in managing adults with the condition and has the potential to improve outcomes .

**Ethics and dissemination** The trial has ethical approval from the UK National Research Ethics Committee

## Strengths and limitations of this study

► DexEnceph will be the first completed randomised controlled trial of corticosteroids in herpes simplex virus encephalitis, examining the utility and safety.

► DexEnceph's primary end point is verbal memory score recorded at 26 weeks after randomisation, this represents the most important neuropsychological damage

► The recruitment target is informed by the recent Enceph-UK programme grant of encephalitis in the UK; the trial is currently open and has recruited 82 patients of a target 90.

► Innovative methods for engaging with recruitment sites have been key to ensuring the success of the study.

(Liverpool Central, REF: 15/NW/0545, 10 August 2015). Protocol V.2.1, July 2019. The results will be published and presented as soon as possible on completion.

**Trial registration numbers** ISRCTN11774734, EUDRACT 2015-001609-16.

## INTRODUCTION

Herpes simplex virus infection (HSV) is the most commonly identified viral cause of encephalitis, inflammation and swelling of the brain caused by a virus or the body's immune system, in the UK as in most western industrialised nations.[1–4] The incidence has been estimated at 1 in 250 000–500 000,[2] with evidence it may be higher.[4] Although a rare disease, HSV encephalitis has a disproportionately large impact due to its devastating long term neuro-psychological sequelae . These can have a marked impact on the quality of life of the patient and their family and high health economic and social costs.[5 6]

Since the introduction of the antiviral drug aciclovir in the 1970s, the mortality of HSV encephalitis has reduced from around 70% to 5.5%–12%.[7–9] However, survivors are commonly left with neurological impairment; less than 20% of patients are able to return to work and 48% are classed as moderate to severely disabled.[10] Even when obvious disabilities have not occurred, families often report personality changes—the person they take home from hospital is simply not the same as the one before the illness.[6 11]

HSV encephalitis can cause a broad range of cognitive impairments, but impaired memory, especially verbal memory, is the most common and likely relates to the viral predilection for the temporal lobe of the brain.[12 13] The verbal memory deficits manifest as difficulties remembering names of objects and people, as well as listening to and recalling spoken information for example in conversations.[14 15] In addition to memory problems, difficulties in processing speed, concentration, language and executive function are also common among survivors of HSV encephalitis, along with fatigue, poor concentration, anxiety and depression.[7 16 17]

The pathogenic mechanisms in HSV encephalitis are not fully understood. The evidence suggests that in addition to direct viral pathogenesis, inflammation of the brain in response to the virus is a key component of the disease process.[18–21] This is supported by the observation that in the cerebrospinal fluid (CSF), higher levels of proinflammatory chemokines, especially monocyte chemotactic protein-1, interferon γ and interleukin 6 (IL-6) are associated with a worse prognosis.[22 23] Poor prognosis is also associated with the extent of inflammation seen on neuroimaging,[24] and the degree of temporal lobe swelling is correlated with the severity of verbal memory impairment.[25] Relapse occurs in about approximately 10% of HSV encephalitis patients, and is sometimes associated with the development of anti-N-Methyl-D Aspartic acid (NMDA) antibodies.[26]

Control of the inflammation in HSV encephalitis may improve outcome, as shown in mouse models of the disease.[27–29] Before the availability of aciclovir, corticosteroids were sometimes used as a treatment in humans with HSV encephalitis,[30 31] and more recently both cerebral oedema on imaging and CSF IL-6 levels were shown to be reduced in patients given corticosteroids.[23] However, because corticosteroids cause immunosuppression, which in theory facilitates increased viral replication, their role is uncertain.[32]

In other brain infections, including bacterial meningitis and tuberculous meningitis the benefit of corticosteroids has been demonstrated in large clinical trials.[33] For HSV encephalitis the potential benefit of using corticosteroid as an adjunct to aciclovir therapy has been suggested from small case series and retrospective comparisons, but there has been no prospective randomised study reported.[34–39] One study, the German trial of Aciclovir and Corticosteroids in Herpes simplex virus Encephalitis (the GACHE trial) was stopped early because of poor recruitment rates.[40] However, there is clearly a need for a study to answer this question. The DexEnceph Study, a randomised controlled trial of dexamethasone in HSV encephalitis, aims to achieve this.

## Trial design

DexEnceph is a pragmatic, multicentre, randomised, controlled, observer-blind trial to determine whether the addition of dexamethasone to standard aciclovir treatment improves clinical outcomes (in particular verbal memory score) for adults with HSV encephalitis. Additionally, neuroimaging and biomarkers are assessed along with detection of HSV in the CSF at 2 weeks after randomisation to monitor for difference in viral replication between the two groups.

## Primary objective

To determine whether a short course of intravenous dexamethasone, in addition to standard care, improves verbal memory score in adults with HSV encephalitis at 26 weeks after treatment compared with standard care alone.

## Secondary objectives

Secondary objectives include the following:

To determine whether dexamethasone therapy has an effect on other neuropsychological, cognitive, clinical, disability and functional outcomes in HSV encephalitis.

To assess the effect of dexamethasone therapy on brain swelling examined by neuroimaging .

To determine whether dexamethasone therapy affects clearance of HSV from the CSF, the emergence of NMDA receptor antibody or causes any changes in transcriptomic and proteomic profiling in the CSF and blood.

A more comprehensive list of measures is detailed in the outcomes section.

## METHODS AND ANALYSIS

DexEnceph is an observer-blind, open-label, prospective, randomised, controlled trial of dexamethasone at 10 mg four times daily for 4 days, versus no dexamethasone, in adults with HSV encephalitis.

## Research setting

The trial is being conducted in up to 45 National Health Service (NHS) trusts, with a recruitment target of 0–2 patients per site per year. A full list of sites involved in DexEnceph can be obtained from www.dexenceph.org. uk. The trail uses Standard Protocol Items: Recommendations for Interventional Trials reporting guidelines[41] (online supplemental file 1).

## Eligibility criteria

### Inclusion criteria

Enrolled patients fulfil all of the following criteria:

1. Suspected encephalitis defined as: new-onset seizure or new focal neurological signs or alteration in consciousness, cognition, personality or behaviour. Personality/behavioural change includes agitation, psychosis,

somnolence, insomnia, catatonia, mood lability, and altered sleep pattern.

2. A positive HSV DNA PCR result from CSF, reported not more than 7 days prior to randomisation.
3. Receiving intravenous aciclovir administered as 10 mg/kg three times daily or at a reduced dose if clinically indicated.
4. Age≥16 years.
5. Written informed consent given by the patient or their legal representative[42]

## Exclusion criteria

Patients are excluded if they have any of the following:

1. Have received oral or injectable corticosteroid therapy in the 30 days prior to the day of entry to the study. This does not apply to topical/ inhaled corticosteroids. (Patients who have received oral or injectable corticosteroid therapy AFTER their admission to hospital will not be excluded from the study if they consent to trial participation).
2. History of hypersensitivity to corticosteroids.
3. Immunosuppression secondary to:
   a. Known HIV infection and CD4 white cell count under 200/mm[3.]
   b. Currently taking biological therapy or other immunosuppressive agents (eg, azathioprine, methotrexate, ciclosporin)
   c. Previous solid organ transplant and currently on immunosuppression.
   d. Previous bone marrow transplant.
   e. Currently undergoing a course of chemotherapy or radiotherapy.
   f. Known primary immunodeficiency syndrome.
   g. Known current haematological malignancy.
4. Pre-existing indwelling ventricular devices.
5. Peptic ulcer disease in the last 6 months, defined as a peptic ulcer seen at endoscopy or an upper gastrointestinal bleed causing a ≥2 unit haemoglobin drop, in the last 6 months.
6. Antiretroviral regimen containing rilpivirine as current treatment (levels of rilpivirne are known to significantly decrease in coadministration with dexamethasone, a switch to a suitable alternative can facilitate trial entry).

## Intervention

Participants are randomised in a 1:1 ratio to dexamethasone four times daily for 4 days alongside standard care, or standard care alone (figure 1). Standard care includes intravenous aciclovir for a minimum of 14 days based on an ideal body weight at 10 mg/kg every 8 hours, unless dose adjustment to account for renal impairment is necessary. Participating clinicians remain free to modify or discontinue the dexamethasone administration or to give alternative treatments at any stage, if this is judged to be in the best interest of the patient.

Participants assigned dexamethasone receive 10 mg equivalent of ordinary ward stock, prescribed by an authorised member of the local study team, given intravenously

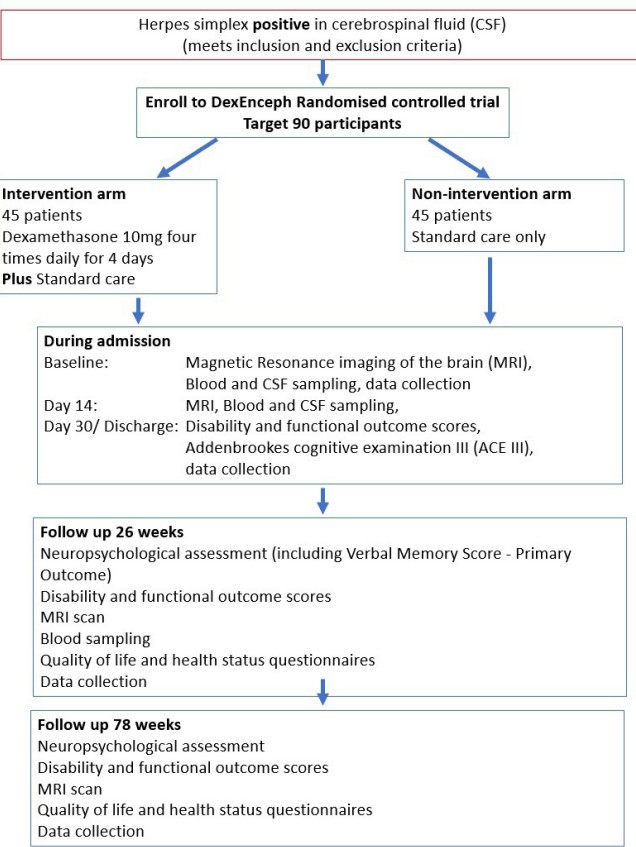

**Figure 1** Schematic design of DexEnceph randomised controlled trial.

four times daily for 4 days (16 doses in total) starting within 24 hours of randomisation.

The University of Liverpool employs a clinical trials unit to be responsible for screening, and monitoring data collection, quality and completeness. As there is a low number of participants to be recruited, the trials unit are able to liaise regularly with each site following randomisation to ensure all follow-up data are collected. Primary outcome is recorded by a centrally employed roving neuropsychologist, who collects the neuropsychological outcomes at 26 and 72 weeks.

## Outcome measures
### Primary outcome

Verbal memory score, determined by the Wechsler Memory Scale fourth edition (WMS-IV) Auditory Memory Index, at 26 weeks after randomisation. Patients that die will be allocated the lowest possible WMS-IV score.

### Secondary outcomes

Other neuropsychological outcome measures (at 26 weeks and 78 weeks after randomisation):
► Verbal memory score, determined by the WMS-IV, Auditory Memory Index, at 78 weeks after randomisation.
► Visual, Immediate and Delayed Memory by Indexes of the WMS-IV, processing speed and working memory

subscales from the Wechsler Adult Intelligence Scale Fourth Edition.
- ► Higher executive function using the Trail Making Test.
- ► Anxiety and Depression symptom levels by the Beck Depression Inventory and Beck Anxiety Inventory.
- ► Subjective cognitive complaints using the Perceived Deficits Questionnaire.

Cognitive Outcome Measures (at discharge or 30 days if still in hospital, 26 weeks and 78 weeks).
- ► Addenbrooke's Cognitive Assessment III.

## Clinical Outcomes
- ► Incidence of epilepsy.
- ► Time to hospital discharge.
- ► Requirement of high-dependency unit or intensive care unit admission up to 30 days after randomisation.
- ► Time taken to be free of ventilatory support for 14 days (if any).
- ► Time to reach maximum recorded Glasgow Coma Scale score.
- ► Survival.

Disability and functional outcomes (at discharge or 30 days if still in hospital, 26 weeks and 78 weeks):
- ► Modified Rankin Score, Barthel Index, Liverpool Outcome Score and Glasgow Outcome Scale Extended.

Imaging Outcomes: Change from Baseline at 2 weeks, 26 weeks and 78 weeks
- ► Temporal lobe volume (as percentage of intracranial volume).
- ► Whole brain volume (as percentage of intracranial volume).
- ► Volume of affected region as seen on fluid-attenuated inversion recovery image (as percentage of intracranial volume).
- ► Volume of affected region as seen on diffusion-weighted image (as percentage of intracranial volume).

## Biomarker outcomes
- ► Transcriptomic and proteomic profiling on blood at convalescence (2 weeks and 26 weeks), compared with acute baselines, and on CSF at 2 weeks compared with acute baseline.
- ► Anti-NMDA receptor antibody testing at 26 weeks.

## Safety outcomes
- ► Proportion of patients with detectable HSV in CSF by PCR at 2 weeks.

Health Status and Quality of Life (at 26 and 78 weeks):
- ► Measured by the EuroQoL-5 Dimension-5 Level quality of life scale and 36 item Short-Form Survey self-completed questionnaires.

## Screening
The majority of potential patients are identified by the local research team through identifying patients with a relevant clinical presentation suspicious of HSV

encephalitis and/or detection of HSV in a CSF sample. A screening log is completed for all potential patients. A strong link with the local laboratory is essential as a key factor in ensuring eligible patients are not missed by the investigative team. Investigators for the local research team include neurologists, infectious disease clinicians, acute medics, microbiologist and virologists.

Because this is a rare disease, most centres will only see 1-2 potential patients a year. Extra measures have therefore been taken to try and maximise recruitment. On identifying a suitable patient, sites are able to contact the trial management team for intensive support via a dedicated telephone hotline, email, or an app. Short videos which explain the trial to patients, families and to healthcare workers also support recruitment. Every month, the trial management group monitor the screening reports of each site for the previous 3 months, to ensure they are actively looking for patients. Lower than expected screening is followed up by the central study team making contact with the study site to review their screening methodology and offer support.

## Randomisation
Participants are randomised using a 24-hour secure web-based programme, which is centrally controlled by the clinical trials research centre. Designated members of the trial team at the site (detailed on the delegation of responsibilities log) are provided with a unique username and password which is required to access the web-based randomisation system. In the event of system failure, the patient can be randomised centrally electronically or through secure envelopes. Each participant is allocated a unique study number (randomisation number), the primary identifier for all the participants in this study.

The neuropsychologist collecting the primary outcome and other outcome assessors such as radiologists and the lead investigators are blinded to randomisation during the trial. Trial participants and local site study teams, as well as the trial manager and trial data manager at the clinical trials unit are aware of what treatments have been allocated. The independent data safety and monitoring committee (IDSMC) and statisticians have access to unblinded data grouped by intervention throughout the trial and make recommendations to the trial's steering committee who would only become unblinded in the event of a serious event.

## Participant timeline
The time schedule for enrolment, interventions and assessments is given in table 1.

## Statistical considerations
### Sample size
The primary outcome variable is verbal memory, assessed as part of WMS-IV. In one published series of adults who survived HSV encephalitis, 19 of 22 had memory impairment evident at follow-up, with verbal memory being most severely affected.[12] In that study the mean (SD) verbal

**Table 1**  Time scale for patients randomised in the DexEnceph study

| Procedures | Screening | Baseline | 2 weeks | Discharge or day 30 of admission (whichever is sooner) | 26 weeks | 78 weeks | Premature discontinuation |
|---|---|---|---|---|---|---|---|
| Signed consent form | X* | X† | | | | | |
| Assessment of eligibility criteria | X | X† | | | | | |
| Review of medical history | | X† | | | | | |
| Review of concomitant medications | | X† | X | X | X | X | X |
| Physical exam | | X | | X | | | |
| Study intervention | | X | | | | | |
| Clinical data collection | | X | | X | X | X | |
| MRI scan | | X‡ | X | | X | X | |
| Research blood testing | | X | X | | X | | |
| Lumbar puncture | | X§ | X | | | | |
| Disability and functional outcomes | | | | X | X | X | |
| Glasgow Coma Scale | | X¶† | X¶ | X¶ | X | X | |
| Addenbrooke's cognitive examination | | | | X | X | X | |
| Neuropsychology assessment, including verbal memory score | | | | | X | X | |
| Health status and quality of life questionnaires | | | | | X | X | |
| Clinical laboratory: haematology, biochemistry | | X** | | | | | |
| Assessment of adverse events | | | (X) | (X) | (X) | (X) | (X) |

(X)—As indicated/appropriate.
*Only applicable when patients are prospectively consented for the randomised controlled trial.
†Procedures required before randomisation.
‡Baseline MRI done for clinical purposes can be done from hospital admission up to 7 days after randomisation.
§Diagnostic lumbar puncture for clinical purposes done prior to randomisation.
¶Recorded prior to randomisation, daily for the first 14 days and then weekly until discharge/30 days (whichever sooner).
**Recording of clinic al laboratory tests done for clinical purposes, not as part of trial.

memory score was 88.9 (18.9) compared with the population mean (SD) of 100.[15] This score can only be assessed in survivors, we estimate approximately 10% of patients in the trial will die before assessment of the primary outcome.[8 43 44] In instances where the death is judged to be associated with encephalitis the verbal memory score is recorded as 40, (the lowest possible value which would be obtained even where a patient recorded no recall of any of the items administered in the memory subtests). Where the cause of death is thought to be independent of having encephalitis, those patients will be recorded as lost to follow-up. Similarly, for patients who are too unwell, due to encephalitis, to undergo the assessment, the score is recorded as 40. Decisions as to whether the reasons for death or non-completion of the measures were due to

encephalitis will be made by an independent committee blinded to dexamethasone allocation. Adjusting the estimate of mean and SD from survivors, to include the 10% of patients that die having the lowest possible value of 40, gives a total population mean of 84.8, with a SD of 23.1. A final sample size of 36 participants per group allows us to detect a clinically meaningful difference of 15.5 on the verbal memory score with 80% power, at a two-sided significance level of 0.05. Allowing for up to 20% dropout gives an initial target sample size of 45 participants per group, for a total of 90.

### Statistical analysis
For the primary outcome, participants are included in the analysis based on the intention-to-treat principle. Verbal

memory score will be compared between groups using linear regression. The model will be adjusted for prespecified variables which are judged to be potentially related to the outcome, including age and admission Glasgow Coma Scale Score. No interim analysis is planned, but there is regular monitoring by the IDSMC.

As there may be some missing primary outcome data due to death, inability to complete the assessment, or loss to follow-up, a sensitivity analysis will be carried out. All randomised patients will be included in this analysis.

For continuous secondary outcome variables, comparisons between groups will be analysed as per the primary outcome. The results for residual viral presence in the CSF at 2 weeks will be reported with a 95% CI for the difference in proportions between groups. Time to event outcomes will be analysed using Kaplan-Meier curves, log rank tests and Cox proportional hazards models. Binary secondary outcomes will be analysed using logistic regression.

### Trial promotion and engagement

Because a previous similar study (the GACHE trial) was stopped early due to poor recruitment rates, we have put particular effort into maximising recruitment. This has included keeping the (NOTE) principal investigators, research nurses and the community engaged in the trial. The trial is being publicised regularly using the Encephalitis Society website and newsletter, social media, and patient journey articles. The Encephalitis Society are playing a key role in providing additional support to the patients and their families aside from their work for the trial. At the annual World Encephalitis Day we have engaged with clinicians, patients, families and the public to raise awareness of encephalitis and the trial, especially through social media, newspapers, radio and television. To promote site engagement, study days are arranged for research teams to attend, along with scheduled research nurse teleconferences to allow ideas on maximising recruitment and updates on trial progress to be shared. Research investigators are invoted to the annual Neurological Infectious Diseases course in Liverpool[45] Sites are also kept updated through our website (www.dexenceph.org.uk), and newsletters. In addition we use an innovative sticker chart, whereby a sticker is sent out to every site each time a patient is recruited anywhere in the country, approximately once a month (figure 2); this helps keep the study at the forefront of investigators minds. In addition, in case the DexEnceph study did not recrut to target a parralele study was set up by French colleagues, using the same protocol, so that data could be pooled if needed.

### Trial closure

The end of the trial is defined to be the date on which data for all participants is frozen and data entry privileges are withdrawn from the trial database. The trial may be closed prematurely by the trial steering committee, on the recommendation of the IDSMC if there is sufficient evidence of risk to patient safety.

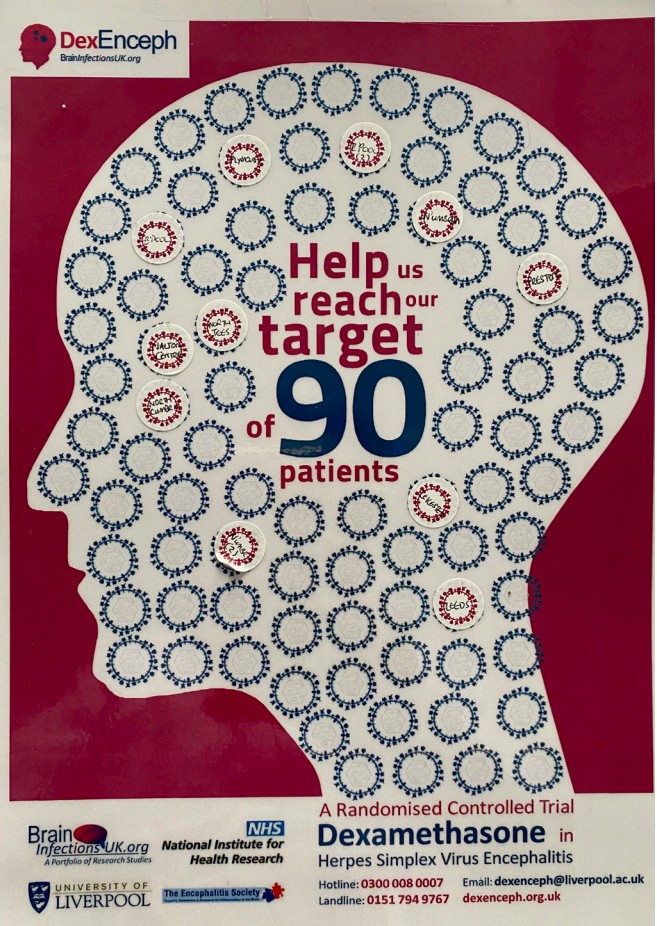

**Figure 2** DexEnceph sticker chart.

### Pharmacovigilance

Oversight of the trial is provided by the trial steering committee, which meets at least annually to review trial progress, safety, and adverse events (AEs). The committee is also informed of any protocol changes by the clinical trial research unit.

The Medicines for Human Use (Clinical Trials) Regulations 2004 (SI 2004/1031) definitions are used for AE, adverse reaction (AR), unexpected AR, serious AE (SAE), serious AR (SAR) or suspected unexpected SAR(SUSAR).

Depending on the nature of the event the reporting procedures below are followed:

1. SAEs occurring up to 30 days after randomisation are reported through an SAE Form (if serious) or in the 30 days/discharge case record forms (CRF) if they are a notable event (positive PCR in CSF at second lumbar puncture, gastrointestinal bleed, hyperglycaemia requiring change in medical management, opportunistic infections, unexpected/severe neuropsychiatric events).
2. SAEs occurring after 30 days from randomisation are monitored through reporting in the CRFs with safety data collected in the 26 weeks and 78 weeks CRFs if serious.

The research investigator at each study site (or designated other) assesses all AEs for seriousness, causality

and severity. The chief investigator (or designated other) assesses all adverse drug reactions for expectedness from known side effects of the use of dexamethasone.[46] All serious ARs, AEs and SUSARs occurring up to 30 days from randomisation (apart from death unless the investigator suspects causality) require reporting to the clinical trials unit, within 24 hours of the site becoming aware of the event. In the case of death of a patient causality will be assessed by the trial steering committee.

The clinical trials unit will notify the Medicines and Healthcare products Regulatory Agency and main research ethics committee of all SUSARs that occur during the study according to the following timelines; fatal and life-threatening within 7 days of notification and non-life-threatening within 15 days. All investigators are informed of all SUSARs occurring throughout the study.

SAEs occurring after 30 days from randomisation are monitored by the clinical trials unit via the 26 weeks and 78 weeks CRFs. These CRFs need to be received at the clinical trials research unit by 4 weeks after the 26 and 6 weeks after 78 weeks time points.

Safety data are provided to the IDSMC, who are responsible for safeguarding the interests of trial participants and assessing the safety of the interventions during the trial; the IDSMC ensures action is taken as needed should they become aware of trends in reported AEs that raise safety concerns.

### Trial funding and financial arrangements

Contractual agreements are in place between the sponsor and collaborating centres that describe financial arrangements. Trial participants are not paid to participate in the trial but are paid travel expenses for the follow-up visits. Payments to sites are made per site initiation but the bulk of payments are made per patient recruitment. Sites receive payment for: clinical time oversight, research nurse time, administrative support, MRI scanning and pharmacy oversight.

### Patient and public involvement

The Encephalitis Society was consulted and provided advice on the design of the trial and the difficulties participants and their families encounter. The chief investigator of the DexEnceph study is the President of the Encephalitis Society. The chief executive of The Encephalitis Society is a coapplicant on the grant application and a coauthor on this paper.

The Encephalitis Society has also provided patient representatives at our trial steering committee and assisted in the production and dissemination of trial promotional materials.

The Encephalitis Society will support publication and dissemination of the trial findings among lay, therapeutic and health professionals through the use of web materials, newsletters and guides as well as at conferences and seminars. All patients and their family/carers will be acknowledged in any outputs from the trial. We also work with The Encephalitis Society on a programme of teaching events and produce guides for healthcare professionals and lay people.

In instances where trial participants and their families have ongoing difficulties the central study team seek help for them through the Encephalitis Society and appropriate specialists for further assistance.

### Ethics and dissemination

The trial falls within the remit of the EU Directive 2001/20/EC, transposed into UK law as the UK Statutory Instrument 2004 No 1031: Medicines for Human Use (Clinical Trials) Regulations 2004 as amended. This trial is registered with the Medicines and Healthcare products Regulatory Agency (MHRA) and granted Clinical Trial Authorisation (CTA). The EUDRACT number for CTA reference is 2015-001609-16. Ethical approval has been obtained from a multicentre research ethics committee familiar with the principals of the Mental Capacity Act 2005 guidance for sites in England and Wales and the Adults with Incapacity Act 2008 for sites in Scotland as the principals are relevant to a clinical trial of investigational medicinal products (CTIMP). Clinical Research Governance approval was given through the Sponsor, The University of Liverpool. The trial protocol was approved by a National Research Ethics Service Committee reference is 2015-001609-16 (Attained 31 March 2016) and underwent independent review at the Research and Development offices at participating sites. This study abides by the principles of the World Medical Association Declaration of Helsinki (1964) and Tokyo (1975), Venice (1983), Hong Kong (1989) and South Africa (1996). Due to the nature of this trial it also abides by the Medicine for Human Use (Clinical Trials) regulations 2004 (S.I.2004:1031) and all following amendments which are incorporated into UK law.

### Informed consent process

In obtaining and documenting informed consent, the investigators adhere to National Institute for Health Research (NIHR) Good Clinical Practice guidelines and the ethical principles derived from the Declaration of Helsinki. Staff delegated by the principal investigator and appropriately trained with experience in obtaining informed consent, discuss the objectives, risks and inconveniences of the trial and the conditions under which it is to be conducted with the patient or if the patient lacks capacity with a legal or professional representative. Trial information documents and points of contact for further information are provided and the potential participants are given adequate time to consider their decision (online supplemental file 2).

As this is a CTIMP, the clinical trial regulations for incapacitated adults are followed (Medicines for Human Use Clinical Trial Regulations 2004 and amendments). When a legal representative has given consent for a patient to participate in the trial and the patient subsequently regains capacity, the research team will provide the patient information sheet and request consent from

the participant. Patients are allowed to withdraw from the study at any point and may request withdrawal of their data collected until this point. Prospective consent can also be obtained prior to a positive PCR result so participants may have adequate time for contemplation.

As suspected encephalitis is a medical emergency, a deferred consent process is used for the collection and retention of some samples as part of routine clinical management. Using emergency deferred consent for samples involves taking additional samples of blood and CSF only if the procedure is being performed for clinical care. If deferred consent has been used, written consent is requested from either the patient or a legal representative as soon as is possible and appropriate, with samples discarded if this is declined. This approach is based on discussions with patients and the public through the Encephalitis Society.

### Data capture methods
Data are stored securely in line with the Data Protection Act 1998. The randomisation system, data capture form and CRF have been designed to optimally protect participant information and to maintain confidentiality. Trial data are captured at local sites using paper CRFs. These are then sent into the clinical trials research unit for data entry into the study-specific database. Completed CRFs are returned to clinical trial research centre within 7 days of completion. A copy of the CRF sent over to the clinical trials research unit is retained at site. CRFs and consent forms are stored separately and securely at all times in dedicated areas of the clinical trials research unit.

CRFs are checked for data quality by the clinical trials research unit in Liverpool responsible for ensuring data collection and storage.

Patients' anonymised and labelled neuroimaging data are put on to discs at site and sent to the clinical trials research unit; the images can also be transferred via the Image Exchange Portal in an encrypted manner. The final dataset will be solely accessible to the central study team at the University of Liverpool for analysis and write up.

### Dissemination
The results of the DexEnceph trial will be published in a high impact journal in a timely manner to present the findings to front-line clinicians. They will be presented at the annual conference organised by the Encephalitis Society and at other meetings. Authorship of the final papers will be determined in accordance with the international committee of medical journal editors' guidelines. The investigators will be involved in the preparation and drafting of the manuscripts. There is no intended use of professional writers.

### DISCUSSION
This protocol describes the design of a randomised controlled trial to examine the role of dexamethasone in the management of patients with HSV encephalitis. HSV encephalitis is a rare sporadic acute disease, and the trial has been designed to take this challenge into account, along with the practicalities of running the trial in a UK National Health Service setting. In particular lessons were learnt from a previous similar European study, the GACHE trial, which was stopped early because of recruitment difficulties. Recruitment to the GACHE trial necessitated patients had focal neurological signs of no longer than 5 days prior to admission, while DexEnceph has less stringent criteria and reflects the diverse ways in which HSV encephalitis may present. DexEnceph has been designed to be both practical and pragmatic, in that patients must be recruited within 7 days of the PCR result becoming available. This allows for occasions where it may take longer to get the PCR performed, and also allows time for patients admitted to district general hospitals, which may not be study centres, to be transferred to larger hospitals which are. DexEnceph also has the advantage in that its recruitment projections were based on preliminary data garnered from the ENCEPH-UK NIHR programme ( www.encephuk.org) and from a multicentre cohort study of encephalitis in England, run by the Health Protection Agency (fore-runner to the Health Protection Agency).[9] These two studies provided direct information on the number of HSV encephalitis patients presenting to UK hospitals. The GACHE study was a double-blind placebo controlled study. Our choice of an open-label observer-blind study, avoided the logistic challenges of ensuring blinded study drug was available across the large number of centres, which might each see only one to two patients per year. We are confident our robust monitoring and trial promotion ensures the majority of eligible patients are recruited. NMDA receptor antibody encephalitis (which is treated with corticosteroids and other immuno-modulatory therapies) is being recognised increasingly as a late complication of HSV encephalitis.[47 48] DexEnceph may also be able examine whether corticosteroids reduce the incidence of this complication. If there is demonstrable efficacy of corticosteroid in improving neuropsychological, imaging and quality of life outcomes, without compromising patient safety the results will be far reaching.

### Collaboration with France
Because we recognised from the start that there may be difficulties keeping to recruitment targets in the DexEnceph study, we worked with colleagues in France to develop a parallel French study (DexEnceph-France). This follows the UK DexEnceph protocol as closely as possible, while being pragmatic about the constraints of a different country's healthcare system. The French trial is based in 10 hospitals with the lead centre being Grenoble Alpes University Hospital and the aim of recruiting 30 patients.

The intention is for the two trials to be analysed separately, with the option of also combining them into an

overall analysis which will give additional power to detect a treatment effect.

## Trial status

The trial was opened in the UK in August 2016. By June 2019 it was running at 45 NHS trusts, and

by Feburary 2020 71 patients had been randomised. However, in March 2020 recruitment to the trial was paused, along with most other NIHR-funded studies, because of the Covid-19 pandemic. However ongoing assessment was completed for patients in the study, and the primary outcome was not missed for a single such patient. Conducting such assessments despite social distancing requirements required some ingenuity on the part of the neuropsychology team. Recuitment to the study reopened in June 2020 (one of the first non-Covid-19 studies to do so) and by June 2021 had recruited 82 (91%) of the target 90 patients. The progress of the trial through the pandemic is a testament to the committment of the research teams around the country, as well as the patients and their families. The trial is currently expected to complete recruitment in January 2022, and to complete followu up for the primary outcome 6 months later. The research team have used the lessons learnt in conducting a randomised controlled acute treatment trial of this rare disease to apply succesfully for funding to study the role of intravenous immunoglobulin (IVIG) in autoimmune encephalitis.If you are interested in seeing whether your hospital could become involved in the Enceph-IG study please visit www.liverpool.ac.uk/encephig, or email: encephig@liverpool.ac.uk

## Author affiliations

[1]Department of Clinical Infection, Medical Microbiology and Immunology, University of Liverpool, Liverpool, UK
[2]Clinical Trials Research Centre, University of Liverpool, Liverpool, UK
[3]PLEASE REMOVE THIS ADDRESS ENTRY, X, X, X
[4]Tropical and Infectious Diseases Unit, Liverpool University Hospitals Foundation Trust, Liverpool, UK
[5]Neurology Department, Alder Hey Children's NHS Foundation Trust, Liverpool, UK
[6]Pharmacy Department, Liverpool University Hospitals NHS Foundation Trust, Liverpool, UK
[7]Department of Biostatistics, Faculty of Health and Life Sciences, University of Liverpool, Liverpool, UK
[8]Deptment of Clinical Neuropsychology, The Walton Centre NHS Foundation Trust, Liverpool, UK
[9]Neuroradiology Department, The Walton Centre NHS Foundation Trust, Liverpool, UK
[10]South London Specialist Virology Centre, King's College Hospital NHS Foundation Trust, London, UK
[11]Division of Neuroscience & Experimental Psychology, University of Manchester, Manchester, UK
[12]The Queen's Medical Research Institute, The University of Edinburgh, Edinburgh, UK
[13]The Encephalitis Society, Malton, North Yorkshire, UK
[14]Service des Maladies Infectieuses et Tropicales, CHU Grenoble Alpes, Grenoble, Rhône-Alpes, France
[15]Department of Neurology, Alder Hey Children's NHS Foundation Trust, Liverpool, UK
[16]REMOVE THI ADDRESS, XXXXX, XXX, XXX
[17]Infectious Diseases Department, University of Grenoble, Grenoble, UK
[18]Department of Neurology, Walton Centre NHS Foundation Trust, Liverpool, UK
[19]National Institute for Health Research Health Protection Research Unit in Emerging and Zoonotic Infections, Institute of Infection Ecology and Veterinary Sciences, University of Liverpool, Liverpool, UK

**Contributors** All authors were consulted and inputted into the article, below lists the particular role within DexEnceph. TW: clinical research fellow, CF: clinical research fellow, KD: trials manager, SD: contributor to trial design and running, MG: Clinical and laboratory biomarkers lead, CH: neuropsychology researcher, RT: trial pharmacist, GB: trial statistician, AR-H: trial statistician, PM: neuropsychology lead, KDas: neuroimaging lead, MZ: virology, LP: neuroimaging, SK: neuroimaging, NR: neuroimaging, AE: encephalitis society chief executive advisor, ST: study coordinator France, RK: clinical lead brain infections UK, JPS: principal investigator France, TS: chief investigator responsible for the trial.

**Funding** This trial is funded by the NIHR Efficacy and Mechanism Evaluation Programme for the Department of Health reference 12/205/28.

**Competing interests** TS is supported by the National Institute for Health Research (NIHR) Health Protection Research Unit in Emerging and Zoonotic Infections (Grant No. IS-HPU-1112-10117), NIHR Global Health Research Group on Brain Infections (No. 17/63/110), and the European Union's Horizon 2020 research and innovation program ZikaPLAN (Preparedness Latin America Network), grant agreement No. 734584.

**Patient consent for publication** Not required.

**Provenance and peer review** Not commissioned; externally peer reviewed.

**ORCID iD**
Thomas Whitfield http://orcid.org/0000-0002-4016-2712

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
