## [Reviewer comments · BMJ Open]

ARTICLE DETAILS

TITLE (PROVISIONAL)	Protocol for DexEnceph; a randomised controlled trial of dexamethasone therapy in adults with Herpes Simplex Virus Encephalitis
AUTHORS	Whitfield, Thomas; Fernandez, Cristina; Davies, Kelly; Defres, Sylviane; Griffiths, Michael; Hooper, Cory; Tangney, Rebecca; Burnside, Girvan; Rosala- Hallas, Anna; Moore, Perry; Das, Kumar; Zuckerman, Mark; Parkes, Laura; Keller, Simon; Roberts, Neil; Easton, Ava; Touati, Saber; Kneen, Rachel; Stahl, JP; Solomon, Tom

VERSION 1 – REVIEW

REVIEWER	Dr Fakir M Amirul Islam Swinburne University of Technology, Melbourne, Australia
REVIEW RETURNED	17-Jul-2020

GENERAL COMMENTS	This is a useful and important study. Very clear information and well written. Just a few minor comments. 1. Line 224: OR seems like odds ratio. I would write 'or'.2. Line 235: Age\geq16 year seems very arbitrary age range. It could be \geq18 to say adults or any age limit. Exclusion criteria seems a very long list. It may limit the recruitment of participants.
--

REVIEWER	Richard Whitley The University of Alabama at Birmingham, United States
REVIEW RETURNED	31-Aug-2020

GENERAL COMMENTS	Protocol for DexEnceph: a randomised controlled trial of dexamethasone therapy in adults with Herpes Simplex Virus Encephalitis The current publication details a clinical trial being performed in the United Kingdom to study the impact of dexamethasone on the neurocognitive outcome of patients with herpes simplex encephalitis when administered acyclovir (standard of care). It is designed to accrue 90 patients who will be randomized 1:1 to receive dexamethasone or placebo. The study has been designed by experts in the field of viral infections of the central nervous system. Anecdotal reports as well as animal model data suggest the benefit of this intervention; however, no clinical trial data from rigorously performed studies are available to support this contention. Hopefully, this clinical trial will resolve a long-standing issue as to the value (or lack thereof) of dexamethasone in the management of herpes simplex encephalitis.
--

	The primary endpoint is neurocognitive outcome, a perfectly appropriate outcome measure. Additional secondary and tertiary outcomes will further define the natural history of treated herpes simplex encephalitis in the era of acyclovir therapy. Patients will be followed for 72 weeks. Standardized tools are used to assess neurocognitive outcome. Inclusion and exclusion criteria are clearly delineated and are certainly appropriate for this clinical trial. Essential to inclusion in the study are clinical findings indicative of herpes simplex encephalitis plus a PCR determination on the cerebrospinal fluid indicating the detection of herpes simplex virus DNA. The biostatistical considerations are well identified, including sample size and analysis plan. The investigators have a clinical trials unit available to them such that detailed follow up of those patients enrolled in the study can be appropriately followed. This is not insignificant as patients will be followed over a 72 week period. The investigators have taken into consideration patients lost to follow up, particularly when death will likely account for 10%. To ensure success, the investigators have aligned with the 'Encephalitis Society' of England to accelerate recruitment as well as enhance clinical trial performance. Further, an identical trial is being performed in France which all will allow for validation of the studies being performed in the United Kingdom. To date 79% of the required patients have been recruited into the clinical trial. Thus, it is highly likely that the investigators will be able to answer a question that has been unresolved in the management of patients with herpes simplex encephalitis for decades. This is an excellent clinical trial that is being performed by experts in the field. We all await the results of the study to answer this very important clinical question.
--	--

VERSION 1 – AUTHOR RESPONSE

Response to reviewer one:

Thank you for your feedback

1. The first change has been adopted.
2. We are unable to change the recommended age limit as this has been set since the trial is in progress. In practice there have been no patients less than 18 years of age.

We agree the exclusion criteria is a little prohibitive in length though we have been able to recruit the over two thirds of participants.

Response to reviewer two:

Paragraph 1: Thank you for your feedback, we are hopeful this study will be able to address this question of whether there is benefit to administering steroids in Herpes simplex virus encephalitis.

Paragraph 2: We agree the primary outcome of neurocognitive dysfunction is selected as providing the best chance of demonstrating any effectiveness.

Paragraph 3: The inclusion criteria and exclusion criteria are designed to identify patients undisputably with herpes encephalitis who have no risks or confounders to why dexamethasone would be detrimental or ineffective.

Paragraph 4: Thank you, the trial is designed with contingency for the difficulties these patients face and expected drop out.

Paragraph 5: We are very pleased to be working with the encephalitis society to help provide support to these patients. The French arm of the study is also highly valued.

Paragraph 6: We are very pleased with the trial progress and believe it will provide an answer to this valuable research question.

Paragraph 7: Thank you